# Challenges of COPD Patients during the COVID-19 Pandemic

**DOI:** 10.3390/pathogens11121484

**Published:** 2022-12-06

**Authors:** Sheng-Wen Sun, Chang Qi, Xian-Zhi Xiong

**Affiliations:** 1Department of Critical Care Medicine, Union Hospital, Tongji Medical College, Huazhong University of Science and Technology, Wuhan 430000, China; 2Department of Respiratory and Critical Care Medicine, Union Hospital, Tongji Medical College, Huazhong University of Science and Technology, Wuhan 430000, China

**Keywords:** COVID-19, COPD, ACE2, precocious differentiation of T cells, immunosenescence

## Abstract

Coronavirus disease 2019 (COVID-19) is a severe systemic infection that is a major threat to healthcare systems worldwide. According to studies, chronic obstructive pulmonary disease (COPD) patients with COVID-19 usually have a high risk of developing severe symptoms and fatality, but limited research has addressed the poor condition of COPD patients during the pandemic. This review focuses on the underlying risk factors including innate immune dysfunction, angiotensin converting enzyme 2 (ACE2) expression, smoking status, precocious differentiation of T lymphocytes and immunosenescence in COPD patients which might account for their poor outcomes during the COVID-19 crisis. Furthermore, we highlight the role of aging of the immune system, which may be the culprit of COVID-19. In brief, we list the challenges of COPD patients in this national pandemic, aiming to provide immune-related considerations to support critical processes in COPD patients during severe acute respiratory syndrome coronavirus 2 (SARS-CoV-2) infection and inspire immune therapy for these patients.

## 1. Introduction

The coronavirus disease 2019 (COVID-19) pandemic caused by severe acute respiratory syndrome coronavirus 2 (SARS-CoV-2) is now spreading worldwide, and new cases continue to arise. As of August13 2022, a total of 585,950,085 confirmed cases and 6,425,422 deaths were reported (https://www.who.int/emergencies/diseases/novel-coronavirus-2019, accessed on 13 August 2022). Severe symptoms and high mortality rates are especially common in the elderly and those with underlying diseases, such as cancer [1], hypertension and diabetes [2], and chronic obstructive pulmonary disease (COPD) [3].

In China, the prevalence of CSOPD in COVID-19 patients is 1.5–2.9% [2,4,5], which is much lower than the prevalence of COPD [6]. Thus, it seems that COPD patients are not susceptible to COVID-19. However, when compared with the general population, COPD patients infected with SARS-CoV-2 generally have a higher mortality and poorer prognosis [7], and often need respiratory support [8]. Through the analysis of 1592 COVID-19 patients, Lippi G et al. found that COPD was significantly correlated with severe COVID-19 (OR = 5.69, 95% CI = 2.49–13.00) [9]. The Global Initiative on Chronic Obstructive Pulmonary Disease (GOLD) also indicates that COPD patients are among the worst affected by COVID-19 (https://goldcopd.org/GOLD-COVID-19-GUIDANCE/, accessed on 5 October 2022).

However, one major question is why COPD patients tend to develop severe COVID-19. Therefore, it is important to examine the impact on COPD patients in this crisis.

This review aims to examine several aspects, including innate immune dysregulation, angiotensin converting enzyme 2 (ACE2) expression, smoking, precocious differentiation of T lymphocytes, and immunosenescence in COPD patients, which may contribute to poor outcomes after infection with SARS-CoV-2.

## 2. Links between COPD and COVID-19 Infection 

COPD is a common disease and is the third leading cause of death, causing a heavy economic burden worldwide. In China, the prevalence of COPD in adults over 40 years old is 13.7% [6]. There may be several mechanisms to explain why COPD patients often develop rapid deterioration among critically ill patients with COVID-19. First, the bronchioles of patients with COPD are inherently narrow, therefore the problem of airway obstruction is aggravated, and the insufficient oxygen supply is worsened during the antiviral response. Second, the lung function of COPD patients becomes poorer when infected with SARS-CoV-2, which facilitates the coinfection of other pathogens, such as opportunistic bacteria and viruses, causing a positive feedback loop of worsening lung function and an uncontrolled inflammatory response. Third, due to the high level of basic inflammation in the COPD population, the threshold for induction of inflammatory storms is reduced. Fourth, excessive reactive oxygen species (ROS) accumulation in COPD patients causes lipid oxidation and DNA damage, which initiates an early inflammatory response, and promotes viral replication and survival [10].

In addition to the above well-known pathophysiological theory, some specific immune features in COPD patients may be the key players that contribute to the fatal outcomes of patients with COVID-19 combined with COPD.

## 3. COVID-19 and Innate Immune Dysregulation

The innate immune system is the first active and general line of defense against viral infections. Spike protein or genomic RNA activates the innate immune response by recruiting neutrophils, macrophages, and monocytes, which drive cytokine responses to SARS-CoV-2 infection [11]. However, innate immune dysfunction in COPD patients may cause initial lung injury. These cells release excessive inflammatory signals, such as reactive oxygen species, IL-6, GM-CSF, matrix metalloproteinases, and chemokines, which contribute to lung destruction and the inflammation cascade, rather than viral clearance. Interferons (IFNs) produced by macrophages are considered the most important cytokines in the antiviral response. However, continuous and excessive IFNs become pathologic [12] and induce lung destruction.

There was a significantly higher percentage of CD14^+^CD16^+^ monocytes in peripheral blood from severe COVID-19 patients than in blood from mild COVID-19 patients, suggesting that pro-inflammatory monocytes might be strongly involved in the pathogenesis of the infection. These cells secrete inflammatory cytokines that contribute to the formation of cytokine storms, including GM-CSF, MCP1, IP-10, and MIP1α [13]. The released cytokines also activate macrophages and neutrophils. Importantly, CD14^+^CD16^+^ monocytes were aberrantly increased in COPD patients. Macrophages differentiate into M1 subsets in response to GM-CSF. In general, there are more M1 macrophages in COPD patients than in healthy subjects [14], and high expression of M1 subsets induces the release of pro-inflammatory cytokines such as TNF-α, IL-1 and IL-6, which causes defective innate immunity and severe pulmonary inflammatory cell infiltration. M1 macrophages also promote the polarization of Th1 and Th17, leading to pathogenic inflammatory cell infiltration. Furthermore, the macrophages in COPD have a lower phagocytic ability and fail to eliminate apoptotic cells, leading to secondary release of pro-inflammatory components.

Overall, the innate immune system of COPD patients has reduced antimicrobial activity, compromised antigen presentation, rendered natural killer cell cytotoxicity defective, and decreased links to the adaptive immune system. When encountered with viruses, pulmonary pro-inflammatory monocytes/macrophages from COPD patients release an excessive and aberrant host cytokine storm [15], resulting in lung immunopathological injury.

## 4. COVID-19 and ACE2 Expression

ACE2 is an important regulatory molecule of the renin-angiotensin system (RAS), mediating anti-inflammatory and vasodilation reactions by catalyzing the conversion of Ang II to Ang (1-7) [16]. Increasing evidence suggests that ACE2 plays a protective role in endothelial and lung function. There is a negative correlation between ACE2 expression and COVID-19 severity [17]. Notably, the loss of ACE2 causes an imbalance in the immune system, inducing amounts of pro-inflammatory cytokine secretions, which contributes to the formation of an inflammation storm [18].

The circulating ACE2 expression in children and young people is significantly higher than that in elderly individuals [19], while the expression in elderly women is higher than in elderly male patients [20]. Similarly, SARS-CoV-2 preferentially infects elderly individuals, especially older men [4]. This difference indicates that the expression of ACE2 may be related to the prevalence rate. However, Cai G. thought that age (comparing people older than 60 years old and those less than 60 years old) and sex did not affect the expression of ACE2 [21]. Single-cell RNA sequencing (scRNA-Seq) results even demonstrated that ACE2 expression in the lung was significantly higher and more widely distributed in males than in females (1.66% vs. 0.41%) [22]. Gene polymorphisms of ACE2 [23] and differences in the methods of detection may explain this apparent discrepancy.

There is currently no exact ACE2 expression pattern in COPD patients. Some studies reported that the expression of ACE2 in epithelial cells from COPD patients was significantly higher than that in healthy controls, and negatively correlated with FEV1% [24]. However, by analyzing lung tissue and bronchoalveolar lavage samples, Li et al. found that the expression of ACE2 in COPD patients and healthy populations has no significant differences [25]. Some scientists even reported that the expression of ACE2 in a COPD rat group was lower than that in the control group, and that ACE2 could attenuate the COPD inflammatory process induced by cigarette smoke, by reducing oxidative stress and inhibiting NF-κB and p38 MAPK pathway activation [26]. The apparent contradiction may be due to different expression patterns in different groups or levels. Large cohort studies are needed to clarify the exact expression of ACE2 in different types of COPD patients. The body’s immune response and viral load, rather than the level of ACE2 expression, may be key to the severity of SARS-CoV-2 infection.

## 5. COVID-19 and Smoking

Cigarette smoking is the main factor in COPD patients, which affects the imbalance of the immune system and causes chronic inflammation [27]. A meta-analysis of COVID-19 patients (*n* = 2002) reported that a persistent smoking history was associated with poor prognosis [28]. Through an analysis of COVID-19 patients in China (*n* = 1085) and the United States (*n* = 6637), it was confirmed that the susceptibility to or severity of COVID-19 was positively correlated with smoking status [29].

The prevalence of COVID-19 patients with a history of smoking is 7.63% [7], and the infection rate of smokers is only 1.4% [30]. Although the incidence of smoking is low, the smokers often present a poor outcome [30]. Smoking can not only increase the expression of IL-6, TNF- α, and IL-17 in the lungs of mice, but also induce the expression of Ang II [31]. Smoking also causes respiratory epithelial cell necrosis, destruction of the mucosal barrier, destruction of cilia, continuous inflammation and viral replication [32].

Recently, some researchers proposed a theory that nicotine could prevent the virus from infecting cells and the immune system from overreacting to the virus, thus blocking the cytokine storm responsible for COVID-19 deaths. Moreover, COVID-19 might result from a disorder of the nicotinic cholinergic system caused by SARS-CoV-2, as nicotine was shown to maintain or restore the function of the cholinergic anti-inflammatory system, and control the release and activity of pro-inflammatory cytokines [33]. However, the damaging effects of other toxic chemicals in cigarette smoke far outweigh the potential effects of nicotine.

Collectively, smokers are potential targets vulnerable to the COVID-19 pandemic. In addition, the increased ACE2 expression caused by smoking, and that in the physiological state, may have different roles in regulating the inflammatory response. However, in any case, we recommend that people quit smoking.

## 6. COVID-19 and Precocious Differentiation of T Cells

A study applying single-cell sequencing of alveolar cells in patients with COVID-19 showed that CD8^+^ T cells in severe patients were in a premature or developmental stage, also called the precocious differentiation stage [34]. Moreover, the phenotypes of the circulating naïve T and memory T cell subgroups of COPD patients are biased towards the aging state, resulting in the immune system tending to cause non-specific inflammation. Our unpublished data also suggested that the peripheral blood of COPD patients has decreased naïve T cells and increased memory T cells, which is positively correlated with the severity of COPD.

As the disease progresses, the number of T lymphocytes decreases. Naïve T cells undergo proliferation and differentiation to restore a stable balance of lymphocytes to cope with viruses. However, this self-stable proliferation will skew and induce the precocious differentiation of T cells in the inflammatory microenvironment. That is, naïve T cells are induced by specific memory T cells through non-apoptotic Fas signaling during cell-to-cell contact, while obtaining the characteristics of maternal memory cells. The characteristics of precocious differentiated lymphocytes are as follows: (1) naïve T cells can expand into memory T cells without overt antigen stimulation [35]; (2) the expression of CCR7 and CD62L on the descendants of naïve T cells is decreased [36]; (3) IL-7 and IL-15 can regulate this self-stable proliferation and mediate selective increases in CD8^+^ lymphocytes along with a decrease in the percentage of CD4^+^ T cells [37,38,39]; (4) KLRG1, an important marker of cell senescence, is highly expressed in descendant memory T cells, thus weakening the specific response to antigens [36]; and (5) granzyme and IFN-γ are highly expressed in naïve T-derived progeny cells, which is thought to enhance host tissue injury and exacerbate inflammation [36]. In general, the antiviral effect is severely weakened in naïve T cell-derived progeny cells, while the ability to cause a cascade of events promoting more severe inflammation and host tissue injury is significantly enhanced.

In view of this information, we hypothesized that precocious differentiation of T cells may be a potentially important mechanism underlying acute deterioration of the COVID-19 patient’s condition. In patients with high degrees of impairment of the immune response, high levels of inflammatory factors (IL-1, IL-2R, and IL-6) can promote precocious differentiation of naïve T cells and trigger the formation of a memory-like T cell with weakened antiviral activity and enhanced ability to promote inflammation and destroy the surrounding tissue. Moreover, the activated T cells in turn secrete a plethora of activating cytokines that can enhance the positive feedback loop that leads to vigorous tissue damage and inflammation. Thus, a high level of inflammatory factors induces the aggravation of the inflammatory storm, which leads to acute deterioration in the patient’s condition after a period of stability.

Therefore, when immunocompromised patients encounter a high viral load, precocious differentiation of T cells is induced, as there are not enough specific T cells available to respond to the virus appropriately, which can partly explain why COPD patients are highly susceptible to and suffer from severe complications.

## 7. COVID-19 and Immunosenescence

Age is an independent risk factor for severe COVID-19 patients [40,41]. The mortality increased among people over 65 years old, with people aged 65–79 accounting for 44% of all deaths, and those aged 80 years and over accounting for 46% (https://www.ecdc.europa.eu/en/publications-data/rapid-risk-assessment-coronavirus-disease-2019-covid-19-pandemic-eighth-update, accessed on 8 October 2022). The difference in morbidity and symptoms between young and elderly individuals may be explained by aging. Aging is a complicated and long hypofunctional process caused by oxidative stress, DNA damage [42], early telomerase inactivation [43], and inflammation [44]. Inflammation is a major feature of COPD, with mild systemic inflammatory responses induced by immune senescence, which may be directly involved in the development of COPD.

The aging of the immune system, also called immunosenescence, generally, but not absolutely, appears with aging. Immunosenescence is a reduction in not only the number of lymphocytes but also the diversity and affinity of T cell receptors (TCRs) and B cell receptors (BCRs) [45]. The characteristics of immune aging are as follows: reduced numbers of naïve T cells in the peripheral circulation, specifically in CD8^+^ T cells, and increased numbers of memory T cells; decreased TCR and BCR diversity; decreased T cell function and increased cytotoxicity [46]; low-grade systemic inflammation that increases with aging [46]; and decreased pathogen recognition, chemotaxis, and phagocytosis of macrophages, natural killer cells, and neutrophils.

Generally, the immunosenescence changes of older men are greater than those of women [46]. Men aged over 65 have higher genomic innate and pro-inflammatory activity and lower adaptive activity, and women have more and better specific response cells, which partly explains why there are obvious sex differences regarding SARS-CoV-2 infection [46]. Theoretically, immunosenescence affects the body’s response to various antigens, delaying or weakening the immune response. Decreased lung function and a reduced lung barrier are related to the aging of the lungs. Furthermore, aging cells are less responsive when encountering new antigen stimulation, due to the lower diversity of TCRs or BCRs and decreased naïve T cells. Thus, elevated exhaustion levels and reduced functional diversity of T cells may predict severe progression in COVID-19 patients [47].

Several strategies to improve immunosenescence have been reported, such as transplanting healthy young people’s T cells into patients to improve the function of T cells, and enhance the antiaging and anti-infection effects of the immune system [48]. Moreover, mesenchymal stem cells can regulate immunity and promote repair, proliferation, and differentiation [49], which are of great significance for reversing immunosenescence.

## 8. Management of COPD during the COVID-19 Pandemic

The morbidity and mortality of COVID-19 patients have been increased in COPD patients during the pandemic. For COPD patients who have not been infected with SARS-CoV-2, restricted social distancing and regular drug use should be the primary principle. It is worth noting that the fever, cough, and dyspnea that occurs in an acute exacerbation of COPD are very similar to the symptoms of severe COVID-19. Generally, acute exacerbation of COPD will result in severe respiratory symptoms in the short-term, while the systemic symptoms of COVID-19 are more severe, often taking approximately 7 days to develop severe respiratory symptoms [50]. Detailed examinations, such as pulmonary CT, routine blood tests, and contact history are helpful in the differential diagnosis of acute exacerbation of COPD and severe COVID-19. Once COVID-19 is diagnosed, blood oxygen saturation levels should be closely monitored, and the use of empiric antimicrobials and supportive therapy may be a good choice to control the occurrence of complications and worse outcomes. If there is evidence of disease progression, glucocorticoids should be considered [51,52].

COPD patients are considered to be at a higher risk of developing severe illness from COVID-19. The vaccine is a safe and effective way to deal with this. Generally speaking, the older you are and the more severe your health conditions are, the more important it is to get vaccinated against COVID-19. GOLD 2023 guidelines suggest that people with COPD should have COVID-19 vaccinations in line with national recommendations (https://goldcopd.org/2023-gold-report/, accessed on 5 October 2022). COPD patients would greatly benefit from the vaccine since it not only protects against COVID-19 but also respiratory failure and death [53]. As to whether vaccinations are helpful in the recovery of COPD patients, there is currently no relevant literature; however, our opinion, COVID-19 vaccines can reduce the risk of being seriously ill or dying, and may protect people against different variants of COVID-19. Information on how well the vaccines can protect people with chronic immune conditions or on how or whether COVID-19 affects people with these conditions differently need to be further investigated.

Doctors should also guide patients to maintain their original treatment and self- management by means of a network platform or telephone consultations to reduce the risk of infection and prevent acute exacerbation. Routine pulmonary function examination and imaging examination can be postponed appropriately [50].

## 9. Conclusions

COPD has emerged as a major risk factor for worse COVID-19 outcomes. Herein, we proposed that innate immune dysregulation, increased ACE2 expression, smoking status, precocious differentiation of T cells, and immunosenescence might be possible mechanisms that cause hyperinflammatory responses and severe complications (Figure 1). Elderly individuals, especially COPD patients with a long-term smoking history, are susceptible to COVID-19 and generally have a poor outcome.

The mechanism of high levels of inflammatory storms and the sudden aggravation of COVID-19 in some patients is worth exploring. The precocious differentiation of T cells may partly explain the mechanism. On the other hand, genetic susceptibility may be involved, and additional studies are needed to prove or dismiss this possibility.

Immunosenescence may play a key role in the infection of COVID-19 patients, and additional studies should clarify this view. Stem cell therapy or adoptive naïve T cells from healthy young people to aged individuals may be an effective way to reverse or alleviate the decline in immune system function caused by immune aging. Whether it be for the treatment of the COVID-19 pandemic or prevention of global pandemics that may occur in the future, research on immunosenescence should be a priority.

## Figures and Tables

**Figure 1 pathogens-11-01484-f001:**
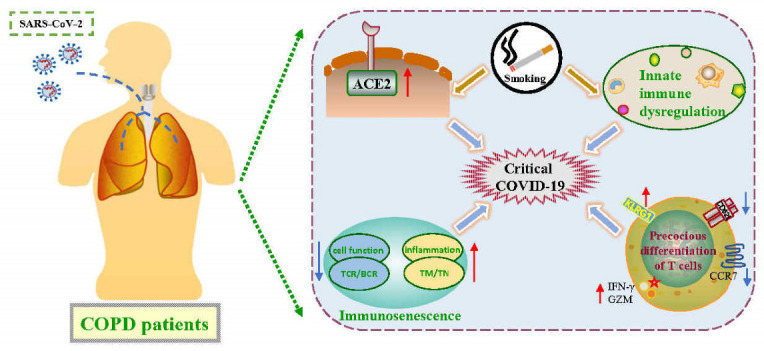
Underlying risk factors in COPD patients, which account for the poor outcomes during the COVID-19 crisis. Smoking induces the expression of ACE2, which mediates SARS-CoV-2 entry and causes innate immune dysfunction characterized by an aberrant host cytokine production and compromised function of immune cells. In addition, precocious differentiation of T lymphocytes was induced when encountered by a high viral load, resulting in high levels of inflammatory factors and host tissue injury. The primary characteristic of immunosenescence is imbalance of memory and naïve T cells, decreased TCR and BCR diversity, decreased function of immune cells, and increased low-grade systemic inflammation. Those factors may explain why COPD patients are highly susceptible to and suffer from severe complications during COVID-19 pandemic. TM: memory T cells; TN: naïve T cells.

## Data Availability

Not applicable.

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
