# Peer review of "Challenges of COPD Patients during the COVID-19 Pandemic"

_pathogens, 2022, doi:10.3390/pathogens11121484_

Round 1
Reviewer 1 Report
COVID-19 is still endangering human being worldwide and those with COPD have a high risk of developing severe symptoms and fatality. It is still an open question why COPD patients tend to develop severe COVID-19. Sun et al. reviewed several risk factors might account for the poor outcomes of COPD patients. And it looks elderly people, especially with a long-term smoking history, might contribute the higher mortality and poorer prognosis. The article should be helpful in understanding the pathogenicity difference of COVID-19 in different populations.
Comments:
1. A major weakness is the lack of quantitative analysis of the contribution of different factors (innate immune dysregulation, increased ACE2 expression, smoking status, precocious differentiation of T cells, and immunosenescence) to serious consequences of COVID-19 on COPD patients. Ranking the importance of these factors will be helpful for the understanding the causes of the severe complications and its treatment.
2. For each mentioned key factor that contribute to the fatal outcomes of patients with COVID-19 combined with COPD, comparing their differences among normal people, patients with COVID-19, patients with COPD and patients with COVID and COPD will help to find the main influencing factors.
3. The manuscript has only one figure. As COVID-19 is still endangering the global human health, if one or more figures are provide for each key factor, it will be helpful for readers without medical background.
Reviewer 2 Report
Authors have written a review on COPD and Covid-19 pandemic. It is a well consolidated article however lacks a perspective on many studies related to COPD patients. Another important point is that in the title of the article it is written Covid-19 epidemic. WHO still hasn't declared it as an epidemic. Authors should also address following concerns:
1. There's conclusive evidence that ACE2 expression is higher in Covid-19 lung tissue patient samples, but based on a few reports the authors have concluded that ACE2 expression and COvid severity are negatively corelated. Its hard to understand the rationale.
2.What is the future perspective of this review
3. Authors should include current therapeutic interventions being used to treat COPD and Covid patients? Including vaccination and drug potentials.
4. What was the impact of vaccination in COPD patients? Was it helpful in recovery?
5. Authors should also include a paragraph on other respiratory viruses and its impact on COPD patients
6. What was the risk assessment of a year outcome between covid and non covid patients? If there's any data authors should include that
Round 2
Reviewer 2 Report
Authors have considerably improved the manuscript. However, the authors still have not corrected the title. They should also include the answer 4 to reviewer's comment in the manuscript.
